# Linear Convergence of Decentralized FedAvg for PL Objectives: The Interpolation Regime

**Shruti Maralappanavar**                                             *mshruti32@gmail.com*
*Department of Electrical, Electronics and Communication*
*IIT Dharwad*
*Dharwad, India*

**Prashant Khanduri**                                          *khanduri.prashant@wayne.edu*
*Department of Computer Science*
*Wayne State University*
*Detroit, MI, USA*

**B. N. Bharath**                                                    *bharathbn@iitdh.ac.in*
*Department of Electrical, Electronics and Communication*
*IIT Dharwad*
*Dharwad, India*

**Reviewed on OpenReview:** *https: // openreview. net/ pdf? id= Og3VxBFhwj*

## Abstract

Federated Learning (FL) is a distributed learning paradigm where multiple clients each having access to a local dataset collaborate to solve a joint problem. Federated Averaging (FedAvg) the algorithm of choice has been widely explored in the classical *server* setting where the server coordinates the information sharing among clients. However, the performance of FedAvg in the *decentralized* setting where only the neighboring clients communicate with each other depending on the network topology is not well understood, especially in the interpolation regime, a common phenomenon observed in modern overparameterized neural networks. In this work, we address this challenge and perform a thorough theoretical performance analysis of FedAvg in the interpolation regime under *decentralized* setting. We consider a class of non-convex functions satisfying the Polyak-Łojasiewicz (PL) inequality, a condition satisfied by overparameterized neural networks. For the first time, we establish that *Decentralized* FedAvg achieves linear convergence rates of $\mathcal{O}(T^2 \log(1/\epsilon))$, where $\epsilon$ is the solution accuracy, and $T$ is the number of local updates at each client. We also extend our analysis to the classical *Server* FedAvg and establish a convergence rate of $\mathcal{O}(\log(1/\epsilon))$ which significantly improves upon the best-known rates for the simpler strongly-convex setting. In contrast to the standard FedAvg analyses, our work does not require bounded heterogeneity and gradient assumptions. Instead, we show that sample-wise (and local) smoothness of the local objectives suffice to capture the effect of heterogeneity. Experiments on multiple real datasets corroborate our theoretical findings.

## 1 Introduction

In the age of Bigdata, Federated Learning (FL) provides machine learning (ML) practitioners with an indispensable tool for solving large-scale learning problems. FL is a distributed machine learning scenario that allows the edge devices to learn a shared model while maintaining the training data at the edge devices (Konečnỳ et al., 2016; McMahan et al., 2017). This avoids the need to share the data with a central server and preserves the privacy of individual clients (edge devices). Assuming a supervised learning setting, where each of the $N$ distinct clients have access to some local data $(\boldsymbol{x}, y) \sim \mathcal{D}_k$ from distribution $\mathcal{D}_k$ for $k \in [N]$

aim to solve:

$$\texttt{FL Problem:} \min_{\boldsymbol{w}\in\mathbb{R}^d} \Phi(\boldsymbol{w}) := \frac{1}{N} \sum_{k=1}^{N} \Phi_k(\boldsymbol{w}),$$

where $\Phi_k(\boldsymbol{w}) := \mathbb{E}_{(\boldsymbol{x},y)\sim\mathcal{D}_k} l_k(f_{\boldsymbol{w}}(\boldsymbol{x}), y)$ is the expected loss at client $k \in [N]$ for the input feature $\boldsymbol{x} \in \mathcal{X}$, and the corresponding label $y \in \mathcal{Y}$. Here, $f_{\boldsymbol{w}}(\boldsymbol{x})$ is the model's output with parameters $\boldsymbol{w} \in \mathbb{R}^d$. In this work, we focus on solving the `FL Problem` in the interpolation regime (cf. Assumption 1), an assumption usually satisfied by overparameterized models (Bassily et al., 2018).

The de-facto standard for solving the above `FL Problem` is the simple Federated Averaging (FedAvg) algorithm (McMahan et al., 2017). In recent years, many works have attempted to characterize the convergence of FedAvg under different settings (Stich, 2018; Li et al., 2019a; Woodworth et al., 2020a; Ma et al., 2018; Yu et al., 2019b). For example, the authors in Stich (2018) show a convergence rate of $\mathcal{O}\left(1/N\epsilon\right)$ for minimizing strongly convex functions while Haddadpour & Mahdavi (2019) establishes similar rates for minimizing functions satisfying Polyak-Lojasiewicz (PL) condition. For minimizing non-convex smooth functions, FedAvg achieves a convergence rate of $\mathcal{O}(1/N\epsilon^2)$ (Karimireddy et al., 2020; Woodworth et al., 2020b), where, $\epsilon$ refers to the desired solution accuracy.

Note that the majority of studies have focused on FedAvg in the classical server-based setting (referred to as *Server* FedAvg), where a central server aggregates information. In this approach, clients compute model updates and send them to the server, potentially leading to communication bottlenecks and delays. Moreover, an attack on the server could compromise the privacy of the aggregated model. Additionally, in many practical learning scenarios, access to a central server may not be feasible. For such cases, the alternative is *Decentralized* FedAvg. In this decentralized setting, global aggregation is replaced by local aggregation, where each client updates its model based on connections with neighboring clients. While the Decentralized FedAvg algorithm has been examined in earlier works (Koloskova et al., 2019; Li et al., 2019b), its linear (fast) convergence has only been established for the strongly convex setting in the interpolation regime (cf. Assumption 1).

In practice, it has been observed that even for non-convex settings *Decentralized* FedAvg converges at a much faster rate compared to the rates demonstrated in these works. To illustrate this fact, in Fig. 1 we plot the behavior of the training loss (on a log scale) as a function of communication rounds for *Decentralized* FedAvg to solve a classification task on MNIST data set utilizing overparameterized models, i.e., in the interpolation regime (for experimental details please see Section 5). It is clear from the plot

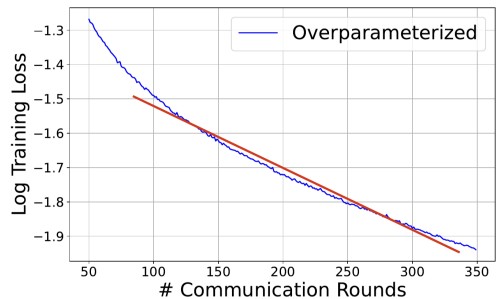

Figure 1: log-training loss vs communication rounds for overparameterized Deep Neural Networks (DNNs) and a simple regression model.

that in the interpolation regime the loss decreases linearly as a function of communication rounds. This suggests that the standard analyses of *Decentralized* FedAvg lack a theoretical foundation to explain its linear convergence.

In this work, we attempt to fill these gaps and perform a thorough theoretical analysis of *Decentralized* FedAvg in the interpolation regime where the local nodes communicate over an undirected graph. Under this setting, we establish the linear convergence of FedAvg for minimizing a class of non-convex functions satisfying the PL inequality. We also extend our analysis to the classical *Server* FedAvg and establish improved guarantees compared to the state-of-the-art. We note that PL inequality plays a key role in the training of overparameterized systems. Specifically, many works have shown that the loss function of an overparameterized neural network satisfies the PL inequality (Bassily et al., 2018; Liu et al., 2020). Furthermore, our analysis reveals that the standard but restrictive assumptions of bounded gradients (Yu et al., 2019b; Stich, 2018; Li et al., 2019a; Koloskova et al., 2020), heterogeneity (Yu et al., 2019a; Woodworth et al., 2020b; Yu et al., 2019a; Wang et al., 2021; Sery et al., 2021), and variance (Woodworth et al., 2020b; Qu et al., 2020) can be avoided while guaranteeing this linear convergence of *Decentralized* FedAvg.

Table 1: Comparison with existing works. Here, SC, C and NC represent Strongly convex, Convex and Non-convex settings, respectively.

| ALGORITHM | CONVERGENCE | EXTRA ASSUMPTIONS | SETTING |
|---|---|---|---|
| Local SGD (Stich, 2018) (s) | $\mathcal{O}\left(1/N\epsilon\right)$ | Bounded gradient | SC |
| Local SGD (Yu et al., 2019b) (s) | $\mathcal{O}\left(1/N\epsilon^2\right)$ | Bounded variance, smoothness | NC |
| Local SGD (Haddadpour et al., 2019) (s) | $\mathcal{O}\left(1/N\epsilon\right)$ | Bounded variance, smoothness | NC |
| FedAvg (Qu et al., 2020) (s) | $\mathcal{O}\left(T\log(1/\epsilon)\right)$ | Bounded Gradient, Bounded Variance | Overparameterized SC |
| Local SGD (Woodworth et al., 2020b) (s) | $\mathcal{O}\left(1/N\epsilon^2\right)$ | Bounded Variance | C |
| Local SGD (Woodworth et al., 2020a) (s) | $\mathcal{O}\left(1/N\epsilon^2\right)$ | Bounded Variance | C |
| PR-SGD (Yu et al., 2019a) (s) | $\mathcal{O}\left(1/N\epsilon^2\right)$ | Bounded Variance | NC |
| FedAvg (Karimireddy et al., 2020) (s) | $\mathcal{O}\left(1/N\epsilon^2\right)$ | Bounded gradient dissimilarity Bounded heterogeneity | NC |
| SGD (Koloskova et al., 2020) (s) | $\mathcal{O}\left(T\log\left(T/\epsilon\right)\right)$ | Smoothness | Overparameterized SC |
| **OUR WORK** (s) | $\mathcal{O}\left(\log\left(1/\epsilon\right)\right)$ | Smoothness | Overparameterized NC |
| NFSGD (Haddadpour & Mahdavi, 2019) | $\mathcal{O}\left(1/N\epsilon^2\right)$ | Bounded local variance | NC |
| DECENTRALIZED SGD (Koloskova et al., 2020) | $\mathcal{O}\left(T\log\left(T/\epsilon\right)\right)$ | Smoothness | Overparameterized SC |
| DECENTRALIZED GD(Song et al.) | $\mathcal{O}\left(\log\left(1/\epsilon\right)\right)$ | Smoothness | Overparameterized NC |
| **OUR WORK** | $\mathcal{O}(T^2\log(1/\epsilon))$ | Smoothness | Overparameterized NC |

**Contributions.**   Here, we list the major contributions of our work:

1. First, we consider the *decentralized* setting where $N$ distributed clients communicate over an undirected graph. We show that to achieve an $\epsilon$-accurate solution *Decentralized* FedAvg requires $R \sim \mathcal{O}\left(T^2\log\left(1/\epsilon\right)\right)$ rounds of communication. We also characterize the effect of the network topology on the performance of *Decentralized* FedAvg.

2. For the classical *Server* FedAvg also we establish linear convergence rates of $R \sim \mathcal{O}\left(\log\left(1/\epsilon\right)\right)$. Compared to the best-known result in Koloskova et al. (2020) where a simpler strongly-convex problem is considered (see section C for details), we get improved rates independent of the number of local updates $T$ (cf. Table 1 and Section 1.1 for a comparison).

3. Our theoretical results do not require assumptions on the boundedness of heterogeneity, gradients, and stochastic variance. We show that sample-wise smoothness of the stochastic loss functions suffices to capture the effect of data heterogeneity among different clients while avoiding the need to impose the restrictive bounded gradient and variance assumptions.

4. Finally, we present experimental results on various data sets such as CIFAR-10, Shakespeare, MNIST, and FMNIST, and corroborate our theoretical findings under the *decentralized* settings. Through our experiments, we show that an optimal number of local updates $T_{th}$ exists and increasing $T$ beyond $T_{th}$ hurts the performance of FedAvg in both *server* and *decentralized* settings.

## 1.1   Related Work

After the introduction of the FedAvg (McMahan et al., 2017), multiple works have analyzed the convergence of FedAvg in the server setting with homogeneous data, i.e., when the data is i.i.d across clients (see (Stich, 2018; Wang & Joshi, 2018; Khaled et al., 2019; Yu et al., 2019b; Wang et al., 2019; Yang et al., 2021)). The authors in (Stich, 2018) were the first to obtain a rate of $\mathcal{O}(1/N\epsilon)$ for strongly convex and smooth problems for achieving an $\epsilon$-accurate solution (cf. Definition 2). Later (Haddadpour et al., 2019; Haddadpour & Mahdavi, 2019) proved a similar result but for non-convex functions satisfying PL inequality. The analysis of FedAvg for the general non-convex settings was first performed in (Yu et al., 2019b) where the authors establish a rate of $\mathcal{O}(1/N\epsilon^2)$ to reach an $\epsilon$-stationary point where the stationarity is measured with respect to the gradient norm of the loss funciton.

There are a few works that have analyzed the performance of Fedvg in *decentralized* settings as well. One of the initial works, (Lian et al., 2017) considered a decentralized parallel SGD (D-PSGD) and provided a convergence rate of $\mathcal{O}(1/N\epsilon^2)$ to an $\epsilon$-stationary point to minimize smooth non-convex functions. Later,

(Haddadpour & Mahdavi, 2019) analyzed the convergence of FedAvg under both *server* and *decentralized* settings with bounded gradient dissimilarity assumption. The authors also established a convergence rate of $\mathcal{O}(1/N\epsilon^2)$ to an $\epsilon$-stationary point to minimize nonconvex functions in both the *server* and *decentralized* settings. The authors in (Yu et al., 2019a) also extended the analysis of Momentum SGD to decentralized networks and established convergence of $\mathcal{O}(1/N\epsilon^2)$ to an $\epsilon$-stationary point to minimize non-convex functions. Recently, the authors in (Song et al.) established a linear convergence rate of $\mathcal{O}(\log(1/\epsilon))$ to an $\epsilon$-accurate solution for a decentralized gradient descent algorithm in the overparameterized regime. But, it is well known that in the deterministic setting distributed algorithms for minimizing PL loss functions are capable of achieving linear convergence (even in the non-interpolated regime). Moreover, for data-centric problems, deterministic algorithms involve computations of large batch gradients during each update, making them impractical for large-scale problems. Further, the local nodes conduct only a single local update. However, in this work, we consider a *stochastic problem* where each agent conducts *multiple local updates* within each communication round. Moreover, it is an open problem to establish if (stochastic) *Decentralized* FedAvg can guarantee linear convergence in the interpolation regime for non-strongly convex losses. All the above works provide a sublinear rate of convergence for *Decentralized* FedAvg, however, as illustrated in Fig. 1, *Decentralized* FedAvg converges at a much faster rate in practice. To understand this behavior of *Decentralized* FedAvg, in this work, we analyze the performance of *Decentralized* FedAvg for minimizing a special class of non-convex functions satisfying PL inequality under the interpolation regime. We note that overparameterized neural networks/systems usually operate in the interpolation regime while their loss functions have been shown to satisfy the PL inequality.

Similarly, the authors in (Koloskova et al., 2020) have also established the linear convergence of FedAvg in the *decentralized* setting for minimizing strongly-convex losses in an overparameterized setting. The above works only focus on the analysis of FedAvg for the strongly-convex objectives in the overparameterized regime while we focus on the more general class of non-convex functions satisfying the PL inequality. Note that for strongly-convex objectives the local clients will all share a unique minimum, this implies that even if there is no communication, FedAvg will converge with the same rate to the local optimal. Importantly, our analysis improves the analyses of (Qu et al., 2020) and (Koloskova et al., 2020), and establishes better dependence on the local updates on the performance of FedAvg (Please see Table 1). Moreover, compared to other works that assume restrictive bounded gradient, heterogeneity, and variance assumptions, we show that such assumptions can be avoided by using a sample-wise smoothness assumption. The authors in (Qu et al., 2020) show linear speedup for strongly convex and convex functions. Table 1 presents a summary of the above discussion.

The *Decentralized* SGD algorithm is extensively studied under various conditions which consists of time-varying network graphs (Koloskova et al., 2020), momentum updates (Lin et al., 2021), asynchronous model updating (Nadiradze et al., 2021). To handle the heterogeneity in data across the clients, various tracking algorithms such as gradient tracking, model tracking (Yue Liu & Stich, 2024; Aketi et al., 2024) and momentum tracking (Takezawa et al., 2023) have also been proposed. In separate line of works, the authors in (Zhu et al., 2023b;a) prove generalization guarantees for *Decentralized* SGD algorithm. The authors in (Nadiradze et al., 2021) along with asynchronous updates consider quantization and multiple local steps. Some recent works such as (Beznosikov et al., 2022) have looked at the stochastic extragradient method with time varying networks for the decentralized methods.

**Notations:** We use bold small letters to denote vectors and capital bold letters for matrices. We denote the expected value of a random variable $X$ by $\mathbb{E}[X]$. We denote $l_2$-norm by $\|\cdot\|_2$, Frobenius norm by $\|\cdot\|_F$ and operator norm by $\|\cdot\|_{op}$. $\langle\cdot,\cdot\rangle$ denotes the inner product. The cardinality of any set $\mathcal{B}$ is represented by $|\mathcal{B}|$. We use the standard notation $\mathcal{O}(n)$ to denote the order of $n$. For a vector-valued function $\Phi(\boldsymbol{w})$, the gradient is denoted by $\nabla\Phi(\boldsymbol{w}) \in \mathbb{R}^d$, and the Hessian is denoted by $\nabla^2\Phi(\boldsymbol{w}) \in \mathbb{R}^{d\times d}$. We use $\mathbf{1}$ to represent a column vector with all ones. We use $[N]$ to denote the set $\{1,\ldots,N\}$.

## 2 The *Decentralized* FedAvg Algorithm

In many practical settings, the central server is absent, and the clients are required to communicate and update the model weights in a decentralized manner while communicating with only the neighboring nodes. In this work, we consider such a setting and study *Decentralized* FedAvg, an extension of FedAvg to the decentralized setting. The *decentralized* setting consists of $N$ distributed edge devices which are represented using a connectivity graph $\mathcal{G} \in (\mathcal{V}, \mathcal{E})$. Here, $\mathcal{V} \in [N]$ is the vertex set or clients, and $\mathcal{E} \subseteq \{\mathcal{V} \times \mathcal{V}\}$ represents the edges of the graph. Any edge $(i, j) \in \mathcal{E}$ represents a connection between node $i$ and $j$. Further, the connections are represented using mixing matrix $P = [p_{i,k}] \in \mathbb{R}^{N \times N}$, where $p_{i,k} = 0$ if there is no edge between node $i$ and $k$ i.e., $(i, k) \notin \mathcal{E}$, else $p_{k,i} > 0$. Unlike many existing works on *decentralized* settings (Lian et al., 2017), here we consider a general framework where each client performs $T$ rounds of local updates. *Decentralized* FedAvg is presented in **Algorithm** 1 while in the following, we provide an outline:

1. `Initialization`: Each client $k \in [N]$ initializes the model parameters denoted by $\underline{w}_k^0$. See Step 1 of Algorithm 1.

2. `Local updates`: Each client performs $T$ steps of SGD starting from the model parameters obtained by the aggregation of the updates from neighbouring clients. Towards computing the stochastic gradient, each client $k \in [N]$ uniformly randomly samples a batch of data of size $b$ denoted by $\mathcal{B}_k^{r,t}$, and then computes the gradient. The resulting model parameters after $T$ local rounds in the $r$-th global round are denoted by $w_k^{r,T}$ which is sent to all the neighbouring clients of $k$. See Steps 6 and 7 of Algorithm 1. Note that this procedure is similar to the local update step in the FedAvg case.

3. `Aggregation`: In the $r$-th global communication round, each client $k \in [N]$ computes a local average of the model parameters received by its neighbors. The aggregate model is denoted by $\underline{w}_k^r$. The steps (2) and (3) above are repeated for $R$ rounds. See Steps 11, 12 of Algorithm 1.

### 2.1 Assumptions

In this subsection, we present the assumptions and definitions used in the analysis of the *Decentralized* FedAvg algorithm.

**Definition 1.** *(L-Smoothness): The function $\Phi(u)$ is said to be $L$ smooth if there exists a constant $L > 0$ such that $\|\nabla\Phi(u_1) - \nabla\Phi(u_2)\|_2 \leq L\|u_1 - u_2\|_2$ for any $u_1, u_2 \in \mathbb{R}^d$. Note that this further implies that $\Phi(u_1) \leq \Phi(u_2) + \langle \nabla\Phi(u_2), u_1 - u_2 \rangle + \frac{L}{2}\|u_1 - u_2\|^2$ for any $u_1, u_2 \in \mathbb{R}^d$.*

**Definition 2.** *($\epsilon$-accurate solution): A stochastic algorithm is said to achieve an $\epsilon$-accurate solution in $r$ rounds if $\mathbb{E}[\Phi(w^r) - \Phi(w^*)] \leq \epsilon$, where the expectation is taken over the stochasticity of the algorithm and $w^* \in \arg\min_{w \in \mathbb{R}^d} \Phi(w)$.*

**Assumption 1.** *(Interpolation): We say that the model parameters is operating in the interpolation regime if there exists a $w \in \mathbb{R}^d$ such that the per sample loss $\Phi_{k,j}(w) = 0$ for all samples $j \in [b]$.*

**Assumption 2.** *(PL inequality): The joint objective $\Phi(v)$ satisfies the PL inequality, i.e., $\|\nabla\Phi(v)\|^2 \geq \mu\Phi(v)$ for some $\mu > 0$ and for all $v \in \mathbb{R}^d$. Further, the local loss functions $\Phi_k(v)$ for all $k = 1, 2, \ldots, N$ are also assumed to satisfy the PL inequality, henceforth referred to as local PL inequality, i.e., $\|\nabla\Phi_k(v)\|^2 \geq \mu_k\Phi_k(v)$ for some $\mu_k > 0$ and for all $v \in \mathbb{R}^d$.*

**Assumption 3.** *(Sample-wise, Local and Global smoothness): The functions $\Phi_{k,j}(\cdot)$ for all $j \in [b], k \in [N]$ are assumed to be $l_{k,j}$-smooth. The local functions $\Phi_k(\cdot)$ for all $k \in [N]$ are assumed to be $L_k$-smooth. The above assumptions imply $\|\nabla\Phi_{k,j}(v)\|^2 \leq 2l_{k,j}\Phi_{k,j}(v)$ and $\|\nabla\Phi_k(v)\|^2 \leq 2L_k\Phi_k(v)$ for all $k \in [N]$ and $j \in [b]$. We also assume the global loss $\Phi(\cdot)$ to be $L$-smooth.*

**Assumption 4.** *(Unbiasedness): We assume that the stochastic samples of the gradient and the loss function at each client $k \in [N]$ are unbiased, i.e., $\mathbb{E}[\nabla\Phi_{k,j}(w)] = \nabla\Phi_k(w)$ and $\mathbb{E}[\Phi_{k,j}(w)] = \Phi_k(w)$ for any $j \in [b]$ and $w \in \mathbb{R}^d$.*

Most of the above assumptions, including interpolation, PL inequality, and smoothness are standard assumptions made in various works in the past (Bassily et al., 2018; Karimi et al., 2016; Ma et al., 2018;

---

**Algorithm 1** *Decentralized* FedAvg

---
1: **Initialize** $\{\boldsymbol{w}_k^{0,0} = \underline{\boldsymbol{w}}_k^0\}$, $\boldsymbol{w}_k \in \mathbb{R}^d$ for $k \in [N]$
2: **for** $r = 0, 1, \ldots, R - 1$ **do**
3:    **Initialize** $\underline{\boldsymbol{w}}_k^r$ at device $k \in [N]$
4:    **for** $t = 0, 1, \ldots, T - 1$ **do**
5:      **for** devices $k \in [N]$ **do**
6:        **Sample a batch** $\mathcal{B}_k^{r,t}$ and $|\mathcal{B}_k^{r,t}| = b$
7:        **SGD step on** $\boldsymbol{w}_k^{r,t}$ for $k \in [N]$: $\boldsymbol{w}_k^{r,t+1} = \boldsymbol{w}_k^{r,t} - \frac{\eta}{b} \sum_{j \in \mathcal{B}_k^{r,t}} \nabla \Phi_{k,j}\left(\boldsymbol{w}_k^{r,t}\right)$
8:      **end for**
9:    **end for**
10:    **Receive** $\boldsymbol{w}_k^{r,T}$ from clients $k \in [N]$
11:    **Aggregation step :** $\underline{\boldsymbol{w}}_k^{r+1} = \sum_{i \in \mathcal{N}_k} p_{k,i} \boldsymbol{w}_i^{r,T}$, for $k \in [N]$
12: **end for**

---

Nguyen & Mondelli, 2020). For example, the authors in (Bassily et al., 2018; Liu et al., 2022a; Karimi et al., 2016; Haddadpour et al., 2019) assume PL inequality along with sample-wise smoothness to prove linear convergence of FedAvg in the interpolation regime. It is also important to note that the overparameterized systems satisfy PL inequality (Liu et al., 2020; Nguyen & Mondelli, 2020; Nguyen et al., 2021; Allen-Zhu et al., 2019; Liu et al., 2022a), and hence plays a crucial role in the analysis of overparameterized systems (Liu et al., 2022a). Moreover, we note that the assumption of sample-wise smoothness is not very stringent since any neural network with a smooth activation function satisfies this assumption.

## 3   Convergence of *Decentralized* FedAvg

In this section, we prove that the *Decentralized* FedAvg algorithm converges linearly to the global optimum for any smooth non-convex function satisfying PL inequality in the interpolation regime. Compared to the classical *Server* FedAvg analyses this problem poses several challenges. In particular, unlike *Server* FedAvg in the *decentralized* setting each client has access to only parameters from its neighbours. This implies that for *Decentralized* FedAvg we need to handle two drift terms, namely *local drift* and the *global drift*. Local drift refers to the update at each client drifting away from the average obtained from the neighboring clients while the global drift refers to the average obtained from the neighboring clients drifting away from the global average. These two equations are coupled, and hence we use the Lyapunov based approach to show that both drift as well as the loss go down linearly. In addition to the Assumptions 1-4, our analysis also relies on the following assumption on the mixing matrix (Koloskova et al., 2020).

**Assumption 5.** *$P$ is symmetric, i.e., $P = P^T$, and doubly stochastic, i.e., $P\mathbf{1} = \mathbf{1}$, $\mathbf{1}^T P = \mathbf{1}^T$.*

The above assumption covers all networks that are symmetric, for example, fully connected graph, ring graph, (Hua et al., 2022), etc. In the following, we provide the main result for the *Decentralized* FedAvg. The details of the proof are presented in Sec. B of the Appendix.

> **Theorem 1 (Convergence of Algorithm 2).** *Under Assumptions 1-5, after $T$ local iterations, choosing $\eta = \mathcal{O}(1/T^2)$, we get*
>
> $$\mathbb{E}[\Phi\left(\underline{\mathbf{w}}^{r+1}\right) + \theta \mathcal{D}_{r+1,0}] \leq \left(1 - \frac{\eta \mu}{16}\right) \mathbb{E}[\Phi\left(\underline{\mathbf{w}}^r\right) + \theta \mathcal{D}_{r,0}].$$

*Proof:* See Sec. B.2 in Appendix and Sec. 3.1 for a proof sketch. □

Theorem 1 establishes linear convergence of FedAvg in the *decentralized* setting. A somewhat related work is Koloskova et al. (2020), where they consider a class of $\mu$-strongly convex functions in the decentralized setting, and showed that a linear convergence rate of $\mathcal{O}(\log(\frac{1}{\epsilon}))$ can be achieved by the decentralized FedAvg algorithm in the interpolation regime. Note that in a strongly convex setting, all clients share the unique minima $\mathbf{x}^* = 0$ due to the interpolation assumption. In this case, clients do not need to communicate since

each client can run multiple local rounds to reach the global minima that minimize the average, and hence making decentralized or collaborative learning vacuous! On the other hand, for the PL setting, multiple local rounds may lead to different optimal points, making the problem more challenging. In order to prove linear convergence, the authors in Koloskova et al. (2020) consider bounding a part of the global drift term. More specifically, they consider $\left\| \bar{\mathbf{x}}^{r+1} - \mathbf{x}^* \right\|$ (Lemma 8), where $\bar{\mathbf{x}}^{r+1}$ is the average of the local updates and $\mathbf{x}^*$ is the global optimal. In our setting, we cannot consider $\mathbf{x}^*$ as there is no unique local or global optimal point. Furthermore, they use the property of strong convexity to bound $\left\| \bar{\mathbf{x}}^{r+1} - \mathbf{x}^* \right\|$; this cannot be done in our setting. Hence, our analysis is quite different from the existing work. See Sec. C for more details. Next, we characterize the sample complexity of *Decentralized* FedAvg.

> **Corollary 1.** *Under Assumptions 1-5, to achieve an $\epsilon$-accurate solution, Algorithm 1 requires $R = \mathcal{O}\left(T^2/\mu \left[\log\left(\mathbb{E}[\Phi\left(\underline{w}^0\right)]/\epsilon\right)\right]\right)$ number of communication rounds.*

*Proof:* It is clear from equation 48 of Theorem 1 that $\eta$ scales as $\frac{\mu}{\zeta_8 T^2}$. Thus, from a scaling point-of-view, using $\eta = \frac{\mu}{\zeta_8 T^2}$ in Theorem 1 and the fact that $(1 - x) \leq e^{-x}$, we get

$$\mathbb{E}[\Phi\left(\underline{w}^{r+1}\right) + \theta \mathcal{D}_{r+1,0}] \leq \exp\left(-\frac{R\mu}{\zeta_8 T^2}\right) \mathbb{E}\left(\Phi\left(\underline{w}^0\right)\right).$$

From above we see that to obtain $\epsilon$ accuracy, we want $\exp\left(-\frac{R\mu}{\zeta_8 T^2}\right) \mathbb{E}\left(\Phi\left(\underline{w}^0\right)\right) \leq \epsilon$. Now rearranging the above, and using the fact that $\mathbb{E}[\Phi\left(\underline{w}^{r+1}\right) + \theta \mathcal{D}_{r+1,0}] \geq \mathbb{E}[\Phi\left(\underline{w}^{r+1}\right)]$ gives us the result in the corollary. $\quad\square$

Corollary 1 shows that even in the *decentralized* setting, FedAvg can achieve linear convergence. More importantly, the sum of the drift and the loss goes to zero linearly with $R$, as opposed to most of the existing work (Sun et al., 2021). Observe from Theorem 1 and the corollary above that an $\epsilon$-accurate solution can be achieved if the number of global communications rounds $R$ scales as $\mathcal{O}(T^2)$. For the strongly convex setting since the clients share a unique minima the impact of the local rounds $T$ on the convergence performance is less severe as shown in (Koloskova et al., 2020). Since the local clients share a unique minimum, which allows the algorithm to converge at the same rate to the local optimal irrespective of the local updates and communication protocol used. We believe that the slightly worse convergence of our analysis is due to the non-convex functions satisfying the PL inequality. Importantly, as a special case of our analysis, for the classical FedAvg algorithm (with a server), we establish a convergence rate of $\mathcal{O}(\log(1/\epsilon))$ which is significantly better than that existing results (see Section 4). Note that one can optimize the number of local rounds that lead to faster convergence. However, this optimization is cumbersome in the *decentralized* setting, and hence the convergence depends on $T$.

**Effect of Network Topology.** The effect of *decentralized* clients is captured through the term involving $\lambda_2$. In order to explain the dependency of $\lambda_2$ on convergence, consider the case of $T = 1$, i.e., FedSGD. In this case, if $\lambda_2 \neq 0$ but closer to 1, then the learning rate is dominated by the term $(1 - \lambda_2)/\text{constant}$. Thus, the learning rate is small in the *decentralized* setting (as opposed to $\lambda_2 = 0$). As a consequence, $(1 - \eta\mu/8)$ is closer to 1 leading to a slower convergence. In the extreme case of $\lambda_2 = 1$, i.e., fully disconnected graph, $\eta = 0$, which leads to divergence, as expected. Later, we perform experiments to show the effect of network topology on the convergence for different network settings.

### 3.1 Proof Sketch of Theorem 1

In contrast to the strongly convex setting of (Koloskova et al., 2020), our setting results in each client sharing different optimal points. As a consequence, the execution of multiple local updates within each communication round leads to the following drifts: (i) local drift: multiple local updates (see line 4 to line 8 of Algorithm 1) differ from the average obtained by the neighboring clients, and (ii) global drift: the local average obtained by using updates from the neighboring clients differ from the global average, i.e., the average obtained from all the clients. This implies that we need to control both global and local drifts. We handle this challenge by bounding the loss in terms of the drift term that captures both local and global drifts as mentioned in the Lemma below. We start by proving an upper bound on the average loss $\mathbb{E}\left[\Phi\left(\underline{w}^{r+1}\right)\right]$ in terms of the loss $\Phi\left(\underline{w}^r\right)$ in the $r$-th communication round, and the drift $\mathcal{D}_{r,0}$, as shown in

the following Lemma.

> **Lemma 1.** *The average loss is bounded in terms of the drift as follows*
>
> $$\mathbb{E}\left[\Phi\left(\underline{\boldsymbol{w}}^{r+1}\right)\right] \leq \left(1 - \frac{\eta\mu}{8}\right)\Phi\left(\underline{\boldsymbol{w}}^r\right) + \frac{6\eta L^2 T}{N}\mathcal{D}_{r,0}, \tag{1}$$
>
> *where the drift $\mathcal{D}_{r,0} := \sum_{k=1}^{N}\mathbb{E}\left\|\underline{\boldsymbol{w}}_k^{r,0} - \underline{\boldsymbol{w}}^{r,0}\right\|^2$, and $\eta$ is chosen according to equation 48.*

*Proof:* See Sec. B.2. □

It is easy to see from Lemma 1 that we can obtain the convergence result established in Theorem 1 provided the drift term on the right hand side of equation 1 is bounded in terms of loss. Towards this, first, we bound the drift term which depends on the average loss, leading to two coupled equations (see equation 1 and equation 2). We construct a single equation that is a linear combination of the two coupled equations and show that the linear combination goes to zero exponentially, leading to linear convergence of both drift as well as the loss function. In the following lemma, we provide a recursion of the drift in terms of the average loss and the past drift.

> **Lemma 2.** *The drift is bounded in terms of $\Phi\left(\underline{\boldsymbol{w}}^{r-1,0}\right)$ as follows*
>
> $$\mathcal{D}_{r,0} \leq \left(\left(1 + \frac{1}{\psi}\right)\lambda_2^2 + \eta^2\lambda_2^4\beta T^2 L_{max}^2\right)\mathcal{D}_{r-1,0} + 2\eta^2\lambda_2^4\beta T^2 L_{max}N\mathbb{E}\left[\Phi\left(\underline{\boldsymbol{w}}^{r-1,0}\right)\right], \tag{2}$$
>
> *where $\beta := \frac{4l_{max}(1+\psi)}{\mu_{min}}$.*

*Proof:* See Sec. B.3. □

Next, our task is to show that the recursion in equation 1 and equation 2 satisfy a bound of the form $\Phi\left(\underline{\boldsymbol{w}}^r\right) + \theta\mathcal{D}_{r,0} \leq \upsilon^r \times \left(\Phi\left(\underline{\boldsymbol{w}}^0\right) + \theta\Phi\left(\underline{\boldsymbol{w}}^0\right)\right)$, for some $\theta > 0$ and $\upsilon \in (0,1)$, which is the desired result. In summary, we prove the bounds on the loss (in terms of drift) in Lemma 1 and the drift (in terms of the loss) in Lemma 2. Note that proving the above bound involves carefully handling the drift due to multiple local rounds. Using the above results, we constructed a Lypunov function in terms of drift and loss for some $\theta > 0$, and prove the following bound by appropriately choosing the learning rate $\eta$ and constants:

> **Lemma 3.** *By choosing $\eta$ as in Theorem 1 for some $\theta > 0$, we obtain the following*
>
> $$\Phi\left(\underline{\boldsymbol{w}}^{r+1}\right) + \theta\mathcal{D}_{r+1,0} \leq \left(1 - \frac{\eta\mu}{16}\right)\left(\Phi\left(\underline{\boldsymbol{w}}^r\right) + \theta\mathcal{D}_{r,0}\right). \tag{3}$$

*Proof:* See Sec. B.4. □

The above Lemma shows that the drift and the loss go down to zero exponentially fast (linear convergence), and results in Theorem 1. The above approach is quite different from the existing work (Koloskova et al., 2020).

## 4 Server Setting: FedAvg with Improved Rates

In this section, we show that our analysis specialized to the *server* setting, i.e., $\lambda_2 = 0$ enables us to show that there exists an optimal number of local rounds $T$ that maximizes the rate of convergence. In particular, we provide a closed form solution to the optimum number of local rounds, and show that an improved rate of $\mathcal{O}(\log(1/\epsilon))$ can be achieved as opposed to the existing work (Koloskova et al., 2020) where they show a convergence rate of $\mathcal{O}(T\log(T/\epsilon))$. Further, the same observation is made empirically in both *server* and

the *decentralized* settings (see Sec. 5).

**Optimal local rounds:** Consider the equation for drift from Lemma 2

$$\mathcal{D}_{r,0} \leq \left(\left(1 + \frac{1}{\psi}\right)\lambda_2^2 + \eta^2\lambda_2^4\beta T^2 L_{max}^2\right)\mathcal{D}_{r-1,0} + 2\eta^2\lambda_2^4\beta T^2 L_{max}N\mathbb{E}\left[\Phi\left(\underline{\boldsymbol{w}}^{r-1,0}\right)\right], \tag{4}$$

and a bound on the loss function from equation 35 of Sec. B.2

$$\mathbb{E}\left[\Phi\left(\underline{\boldsymbol{w}}^{r+1}\right)\right] \leq \mathbb{E}\left[\left(\left(1 - \frac{\eta\mu}{4}\right)^T + \frac{64\eta^4 T^3 l_{max}LL_{max}}{\mu_{min}}\right)\Phi\left(\underline{\boldsymbol{w}}^r\right) + \frac{4\eta T L^2}{N}\left\|(Q - P)W^{r,0}\right\|_F^2\right.$$
$$\left. + \frac{2\eta T L^2}{N}\left[\left(\frac{16l_{max}\eta^2 T^2 L_{max}^2}{\mu_{min}} + 4\lambda_2^2\eta^2\gamma L_{max}^2 T^2\right)\mathcal{D}_{r,0} + 4\eta^2\gamma T^2\lambda_2^2\left\|\partial\Phi\left(\underline{W}^{r,0}\right)\right\|_F^2\right]\right] \tag{5}$$

It is evident from the above equations that the drift increases with $T$, as expected. However, a part of the expression in the average loss decreases with $T$ (more specifically, the term $\left(1 - \frac{\eta\mu}{4}\right)^T$) while the other terms increase with $T$. In principle, one should be able to characterize the optimal $T$. However, the above is a complicated expression to optimize with respect to $T$. To get more insights into the effect of $T$, in the following, we look at the *server* setting, which is a special case of our framework. The *server* setting consists of the central server which coordinates the information sharing among participating clients. We obtain the *server* setting by making the second largest eigenvalue of the mixing matrix, i.e., $\lambda_2 = 0$ in the *decentralized* case. Now using the fact that $\lambda_2 = 0$ in equation 4 and equation 5 lead to $\mathcal{D}_{r,0} = 0$ and

$$\mathbb{E}\left[\Phi\left(\underline{\boldsymbol{w}}^{r+1}\right)\right] \leq \left(\left(1 - \frac{\eta\mu}{4}\right)^T + \frac{64\eta^3 T^3 l_{max}L^2 L_{max}}{\mu_{min}}\right)\mathbb{E}\left[\Phi\left(\underline{\boldsymbol{w}}^r\right)\right], \tag{6}$$

respectively. Choosing $\eta \leq \frac{64l_{max}L^2 L_{max}}{\mu_{min}T}$, and utilizing the upper bound $e^{-x} \leq 1 - x + \frac{x^2}{2}$, for all $x \geq 0$ in equation 6, result in

$$\mathbb{E}\left[\Phi\left(\underline{\boldsymbol{w}}^{r+1}\right)\right] \leq \left(1 - \frac{\eta\mu T}{4} + \eta^2 T^2\left(\frac{\mu^2}{8} + 1\right)\right)\mathbb{E}\left[\Phi\left(\underline{\boldsymbol{w}}^r\right)\right].$$

Now we choose $T$ such that the right hand side above is minimized, i.e.,

$$\inf_T\left[1 - \frac{\eta\mu T}{4} + \eta^2 T^2\left(\frac{\mu^2}{8} + 1\right)\right].$$

Note that the above is a convex function. Hence, differentiating the above w.r.t $T$ and equating it to 0, we get

$$T = T_{th} = \frac{\mu}{\eta\left(\mu^2 + 8\right)}.$$

The above analysis leads to the following "faster" convergence rate for the *decentralized* setting:

> **Corollary 2.** *By choosing the number of local updates such that $T = T_{th} = \frac{\mu}{\eta(\mu^2+8)}$, the iterates generated by Algorithm 1 achieve an $\epsilon$ accurate point after $R = \mathcal{O}\left(\frac{8(\mu^2+8)}{\mu^2}\log\left(\frac{\Phi(\underline{\boldsymbol{w}}^0)}{\epsilon}\right)\right)$ communication rounds.*

In the above analysis, we capture the effect of local updates on the performance of the decentralized FedAvg. Specifically, we show that there exists an optimal number of local updates $T$ beyond which the convergence of the algorithm slows down and one needs to choose $T$ carefully to achieve the optimal convergence guarantees. Again, this is the first result establishing linear convergence of FedAvg in the *decentralized* setting when minimizing non-convex functions satisfying PL-inequality in the interpolation regime. In the next section, we present the experimental results.

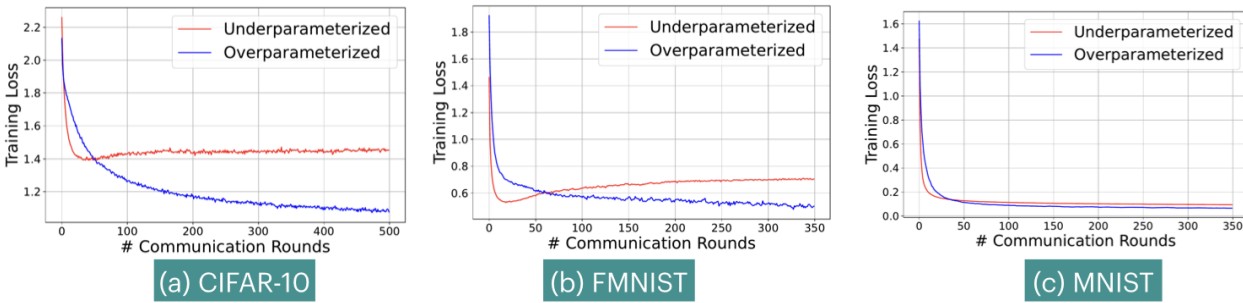

Figure 2: Training loss on different datasets versus the communication rounds for FedAvg in the *decentralized* setting.

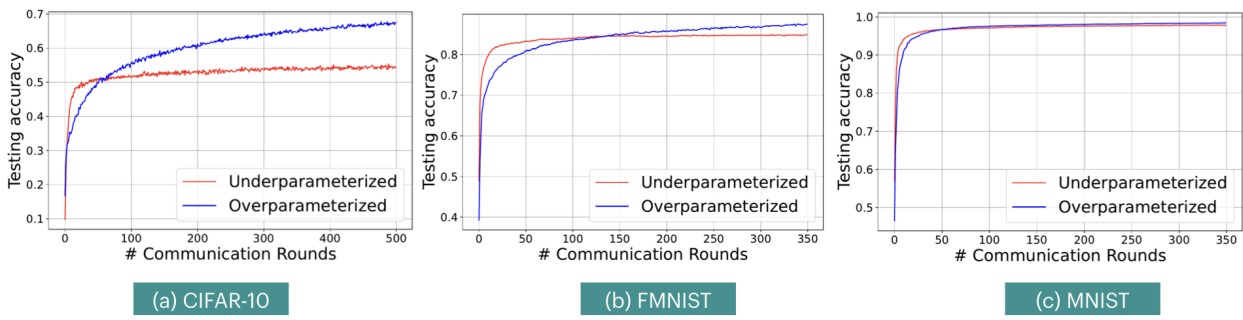

Figure 3: Testing accuracy on different datasets versus the communication rounds for FedAvg in the *decentralized* setting.

## 5 Experimental Evaluation

In this section, we experimentally validate our theoretical findings for the *decentralized* versions of FedAvg. First, we present the experimental setup for various settings.

### 5.1 Setup: *Decentralized* FedAvg

We use 60 edge devices to run the *Decentralized* FedAvg algorithm with multiple local SGD steps and then broadcast the updated model with the nodes connected to it. We consider the image classification tasks on CIFAR-10, MNIST, and FMNIST datasets using an overparameterized simple regression and Deep Neural Network (DNN) models. We have implemented all our experiments on NVIDIA *DGX A*100. The experimental setup consists of the following model and data set:

**Overparameterized regression.** Here we consider a simple regression model with 3 linear layers. There are 231490 trainable parameters with no activation function. We evaluate the performance of *Decentralized* FedAvg algorithm on the MNIST dataset for an image classification task.

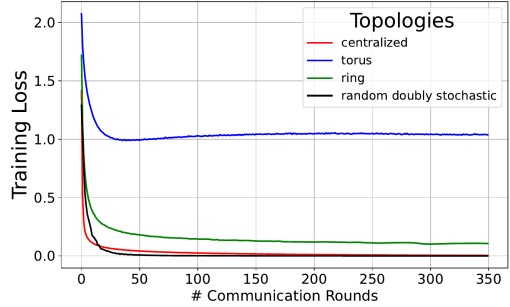

Figure 5: Training loss versus the communication rounds for FedAvg in the *decentralized* setting. Here, random doubly stochastic case has 5 clients while for others we have used 60 clients.

**Deep neural network.** We consider an image classification task under two different settings: underparameterized and overparameterized settings. In this case,

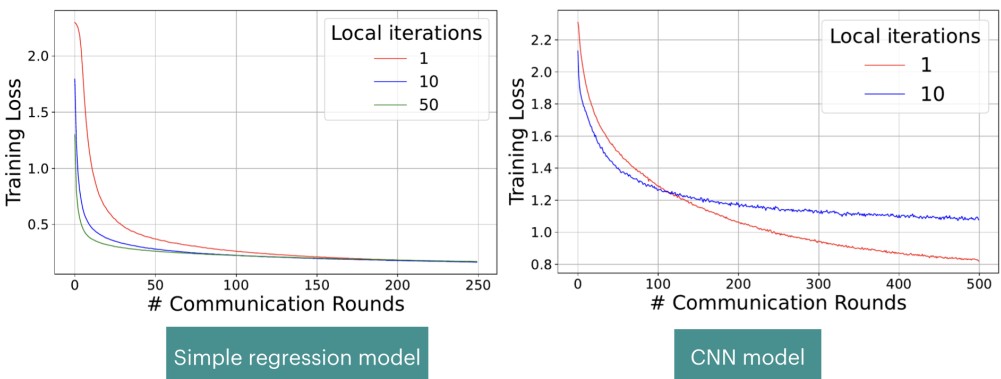

Figure 4: Effect of $T$ on the convergence of *Decentralized* FedAvg for simple regression and CNN model.

each device implements a convolutional neural network (CNN) model. We consider the CIFAR-10, MNIST and FMNIST datasets. In the overparameterized setting, each edge device implements a three hidden layer convolutional neural network (CNN) with 256, 128, and 64 filters followed by three linear layers having 1642849 trainable parameters for CIFAR-10 and two linear layers for MNIST and FMNIST with 1046426 trainable parameters. On the contrary, the underparameterized setting considers a relatively smaller neural network. In this setting, each device implements two hidden layer CNN having 25 and 52 filters followed by two linear layers for CIFAR-10 and one linear layer for MNIST and FMNIST datasets. We set the number of local updates $T = 10$ and pick the tunable learning rate in the range $\eta \in [0.001 : 0.01]$ for CIFAR-10, MNIST, and FMNIST datasets. We consider that each device has 490 training samples and 90 test samples for the CIFAR-10 dataset. On the other hand, for MNIST and FMNIST datasets, 540 samples are used for training and 80 samples are used for testing.

In this setting, we run Algorithm 1 for the following networks (i) ring, (ii) random doubly stochastic, and (iii) torus topologies. For the *Decentralized* FedAvg, we compare (a) the performance of *Decentralized* FedAvg with both underparametrized and overparameterized neural network models, (b) effect of topology on the convergence, and (c) effect of local updates on the convergence. In the following, we provide a detailed experimental results.

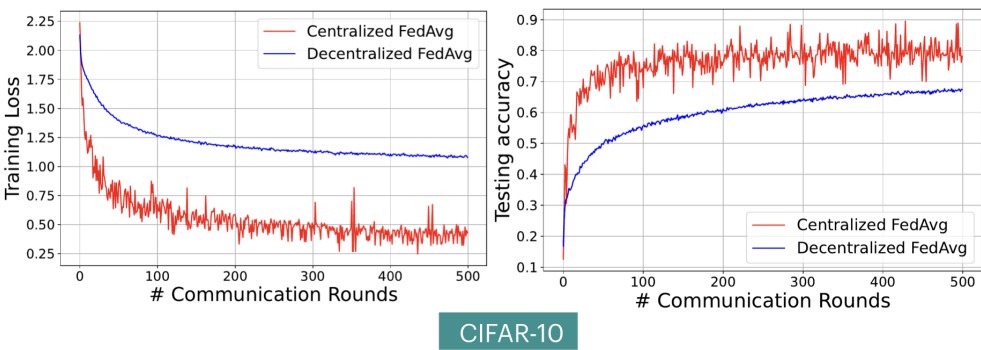

Figure 6: Training loss and Testing accuracy for centralized ($\lambda_2 = 0$) and *Decentralized* FedAvg algorithm with ring topology ($\lambda_2 = 0.33$) on CIFAR-10 dataset versus communication rounds.

## 5.2 Experimental Results for Different Settings

Using the settings described above, here we present experimental results for *Decentralized* FedAvg's, and corroborate various theoretical findings made in this paper:

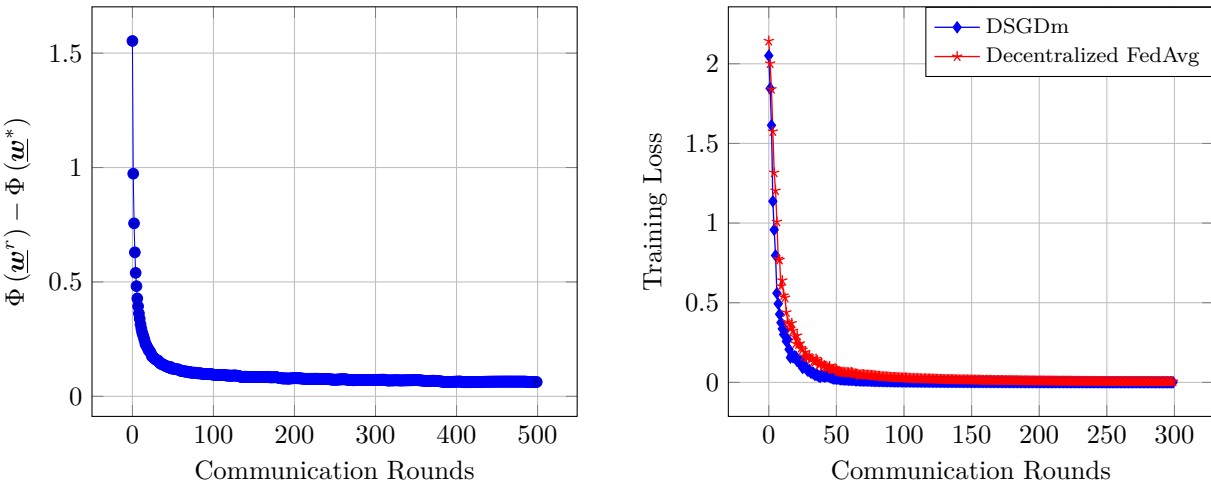

Figure 7: Optimality gap for *Decentralized* FedAvg algorithm with ring topology on MNIST dataset versus communication rounds. (see (a)) and Comparison of the DSGDm and *Decentralized* FedAvg. (see (b)).

1. **Underparameterized versus overparametrized.** Fig. 2 shows the plots of training loss of FedAvg in the *decentralized* setting for underparameterized and overparameterized models on MNIST, FMNIST, CIFAR-10 datasets. As established in Theorem 1, the loss of FedAvg in the *decentralized* setting diminishes rapidly for the overparameterized models compared to the underparameterized models. This is due to the fact that the PL inequality is satisfied for overparameterized systems which helps to reach the global optimum at a linear rate as demonstrated by Theorem 1. Fig. 3 show plots for testing accuracy for FedAvg in the *decentralized* setting. As expected the convergence speed of underparameterized case is slower than the overparameterized case.

2. **Effect of local updates $T$.** Fig. 4 shows plots of the training losses on MNIST dataset for the FedAvg under the *decentralized* setting on the overparameterized regression model and the CNN. From equation 6, we see that as $T$ increases, the convergence speed either decreases or increases depending on the coefficient of $T^3$ in the second term. We capture this phenomenon in Fig. 4. In particular, as $T$ increases, the rate of convergence increases for simple regression model while it decreases/saturates for the CNN based DNN model. One plausible explanation is that the smoothness constants of simple regression is small, and hence results in smaller second term in equation 6. However, for CNN based DNN, the second term dominates, and hence results in slower convergence with $T$.

3. **Effect of optimality gap.** Fig. 7 (a) shows the plot of the optimality gap, i.e., $\Phi\left(\underline{\boldsymbol{w}}^r\right) - \Phi\left(\underline{\boldsymbol{w}}^*\right)$ versus $R$. Here $\boldsymbol{w}^*$ is an approximate optimal solution obtained by running a centralized algorithm for a sufficient number of rounds. As expected the optimality gap decreases exponentially with $R$.

4. **Comparison with the existing work.** Fig. 7 (b) shows the comparison of Decentralized Stochastic Gradient Descent with momentum (DSGDm) (Lin et al., 2021) with the *Decentralized* FedAvg algorithm. We show the training loss versus $R$ for overparameterized CNN model using MNIST dataset in both the cases. We see DSGDm outperforms the *Decentralized* FedAvg algorithm as expected.

5. **Comparison with different topologies in the *decentralized* case.** Fig. 5 shows the training loss versus the communication rounds $R$ for overparameterized CNN model using MNIST dataset with $T = 10$ for four different topologies. Since server topology has $\lambda_2 = 0$, it outperforms the network with ring topology and a random (doubly) stochastic matrix. However, the torus topology does not satisfy the conditions required, i.e., symmetric and doubly stochastic matrix, and hence cannot be used for corroborating our theoretical findings. Nevertheless, we have conducted experiments with torus topology, and Fig. 5 shows that the torus has the worst convergence performance. One reason for this could be that the ring topology has more structure, i.e., it has a symmetric and doubly stochastic mixing matrix $P$ as

opposed to the torus topology. The theoretical analysis of networks with general topology is relegated to our future work.

# 6 Conclusion

In this work, we performed a theoretical analysis of the well known FedAvg algorithm for the class of smooth non-convex overparameterized systems in the interpolation regime. We considered the *decentralized* setting where nodes communicate over an undirected graph. In this regime, it is well know that neural networks with non-convex loss functions typically satisfy an inequality called Polyak-Lojasiewicz (PL) condition. Assuming PL condition, we showed that the FedAvg algorithm achieves linear convergence rate $\mathcal{O}(T^2 \log(1/\epsilon))$, where $\epsilon$ is the desired solution accuracy, and $T$ is the number of local SGD updates at each node. As opposed to the standard analysis of the FedAvg algorithm, we showed that our approach does not require bounded heterogeneity, variance, and gradient assumptions. We captured the heterogeneity in FL training through sample-wise and local smoothness of loss functions. Finally, we carried out experiments on multiple real-world datasets to confirm our theoretical observations.

**Acknowledgements:** This work is in part supported by SERB-CRG (grant number: CRG/2021/007502) and SERB-MATRICS (grant number: MTR/2021/000575). We thank the anonymous reviewers for their helpful comments which have improved the paper.

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
