# OpenReview forum: "Linear Convergence of Decentralized FedAvg for PL Objectives: The Interpolation Regime"
_TMLR — Accepted by TMLR_

### Review · Reviewer_ztBN · 2024-07-28

**Summary Of Contributions:**

The paper shows a linear rate for decentralised SGD where nodes communicate after a number of local iterations. In order to show linear rate, an interpolation assumption is used.

**Audience:**

Yes

**Broader Impact Concerns:**

Nothing in particular.

**Claims And Evidence:**

Yes

**Requested Changes:**

I would like the authors to clarify my questions and modify accordingly the introduction and related work sections in order to make the contribution clear.

**Strengths And Weaknesses:**

Strengths: The paper discusses a topic of interest for the modern machine/deep learning community. Decentralised optimization is crucial in modern learning systems while interpolation is a practical assumption which can explain many phenomena of fast convergence of SGD.

Weaknesses: The paper does not do a good job in explaining the exact contribution compared to related work. I suspect that this contribution is quite limited. This is because interpolation is well-known to be able to provide linear rates for SGD, while decentralised SGD is also very-well studied.

My current understanding is the following and I would like the authors to follow up and clarify the situation:

[The power of interpolation: Understanding the effectiveness of sgd in modern over-parametrized learning] gives linear convergence of SGD under the interpolation assumption.

[ A unified theory of
decentralized sgd with changing topology and local updates] gives (as a side result) linear convergence of decentralised SGD for overparametrized problems in the *strongly convex* case.

[Distributed optimization
for overparameterized problems: Achieving optimal dimension independent communication complexity.] gives linear convergence of deterministic  decentralised GD under the interpolation and PL assumptions.

The current paper gives linear convergence of decentralised SGD under the interpolation and PL assumptions. In addition, the paper includes the scenario where the nodes communicate after T local iterations (but this is usually quite trivial).

The paper also claims to have removed some standard assumptions on bounded variance, heterogeneity etc., but I don't see any of them in [The power of interpolation: Understanding the effectiveness of sgd in modern over-parametrized learning]. Could the authors comment on that?

If my understanding is correct, I think that the paper contains a good to know scenario but the novelty is limited. I leave it to the editor to decide whether this is enough for publication in TMLR.

---

> ### Author Response · Authors · 2024-07-31
> **Response to Reviewer ztBN (Part 1)**
>
> **Your Comment:** Clarify the situation:
> > R1. [The power of interpolation: Understanding the effectiveness of sgd in modern over-parametrized learning] gives linear convergence of SGD under the interpolation assumption.
>
> **Our Response:** We thank the reviewer for the comment. In the following and subsequent responses, we provide a detailed explanation of why our result is not an easy extension of the papers mentioned by the reviewer.
>
>
> The work in [R1] shows linear convergence of single-node **centralized** SGD algorithm. Here, we explicitly state the major challanges of our analyses compared to the **centralized** setting. We note that generalizing the analysis of centralized SGD to decentralized setting, especially, with local updates is not a straight forward task. We list the major differences/challenges below compared to the centralized settings:
>
> - **Local Updates:** Note that in the decentralized setting considered in our work the individual nodes execute multiple local updates. We would first like to point out that the local updates in conjunction with a decentralized topology have received relatively lesser attention by the research community partly because of the challenging nature of the problem. The execution of multiple local updates at each node leads to a phenomenon termed as "client drift", wherein, the locally learned models drift apart from each other because of heterogeneity in the local datasets. In standard, FedAvg or decentralized analyses this drift is controlled using the bounded heterogeneity (or gradient) assumptions, however, it is not clear how to avoid the bounded heterogeneity (or gradients) assumption and still guarantee convergence of FedAvg [1]. This was handled in [R2] but for strongly convex setting. See response related to [R2] below.
>
> - **No heterogeneity/bounded gradient assumption:** The heterogeneity in data across clients needs to be carefully handled in order to obtain linear convergence of FedAvg. We use the smoothness and interpolation assumption to control the heterogeneity and establish for the first time that that we can infact achieve linear convergence for a stochastic problem with each node conducting multiple updates in a decentralized topology.
>
>
> - **Decentralized Topology:** We would also like to point out that compared to the server setting (where a central server coordinates the communication between devices), the decentralized setting presents an additional challange since unlike server setting at the execution of local updates within each communication round the nodes do not have consensus. This implies that in the decentralized setting, we need to control the consensus error in addition to the client drift (see previous point). This poses additional challanges in the analyses.
>
> Reference:
>
> [1] Karimireddy et. al., “Scaffold: Stochastic controlled averaging for federated learning.” ICML, 2020.

---

> ### Author Response · Authors · 2024-07-31
> **Response to Reviewer ztBN (Part 2)**
>
> > **Your Comment:** Clarify the situation:
>
> > R2. [ A unified theory of decentralized sgd with changing topology and local updates] gives (as a side result) linear convergence of decentralised SGD for overparametrized problems in the strongly convex case.
>
> **Our Response:** The authors in [R2] show linear convergence for strongly convex setting. Note that for strongly-convex objectives the local clients will all share a unique minimum, this implies that even if there is no communication, FedAvg will converge with the same rate to the local optimal. In contrast, the local functions in our analysis can converge to different (heterogeneous) local optimum points and, therefore, to achieve consensus (intermittent) communication plays a significant role in our analysis. This makes the analysis more challenging compared to the simple strongly-convex setting. Importantly, through our analysis we show that there exists an optimal choice of local updates beyond which the performance of (decentralized) FedAvg degrades implying that our analysis is considerably more nuanced and achieves better guarantees inspite the fact that we consider a more challenging class of PL objectives **(see page 27 of the updated manuscript)**. In the following, we provide mathematical details on why the analyses in [R2] cannot be applied in our setting.
>
> Note that [R2] considers a class of $\mu$-strongly convex functions. As a consequence, the global objective (sum of strongly convex functions) is also strongly convex. The interpolation assumption ensures the existence of a unique global optimal point $\mathbf x^*$. In order to prove linear convergence, the authors in [R2] consider bounding a part of the consensus term. More specifically, they consider $\lVert\bar{\mathbf x}^{t+1} - \mathbf x^* \rVert$ (Lemma 8 of [R2]), where $\bar{\mathbf x}^{t+1}$ is the average of the local updates and $\mathbf x^*$ is the global optimal. In our setting, we cannot consider $\mathbf x^*$ as the local clients do not have a unique minimum. Furthermore, they use the property of strong convexity to bound $\lVert\bar{\mathbf x}^{t+1} - \mathbf x^* \rVert$; this cannot be done in our setting. Hence, we cannot use the approach taken in [R2]. Therefore, we prove the following bounds on the loss (in terms of drift) and the drift (in terms of the loss) $\mathcal{D}_{r,0}=\sum\_{k=1}^{N}\mathbb{E}\lVert\underline{\mathbf w}\_{k}^{r,0}-\underline{\mathbf w}^{r,0}\rVert^{2}$, as shown in the following theorems.
>
>
> **Theorem:** The average loss is bounded in terms of the drift as follows
> $$\mathbb{E}\left[\Phi\left(\underline{\mathbf w}^{r+1}\right)\right] \leq \left(1-\frac{\eta \mu}{8}\right) \Phi\left(\underline{\mathbf w}^{r}\right)+\frac{6 \eta L^2T}{N} \mathcal{D}_{r,0}.$$
>
> **Theorem:** The drift is bounded in terms of loss as follows
> $$\mathcal{D}_{r,0} \leq
> \left(\left(1+\frac{1}{\psi}\right)\lambda\_2^2 + \eta^{2} \lambda\_{2}^{4}\beta T^2L\_{max}^2\right)\mathcal{D}\_{r-1,0} + 2\eta^{2} \lambda\_{2}^{4}\beta T^2 L\_{max}N \mathbb{E}\left[\Phi \left(\underline{\mathbf w}^{r-1, 0}\right)\right]$$
>
> The parameters above are defined in the paper. Note that proving the above bound involves carefully handling the drift due to multiple local rounds, as apposed to [R2]. Using the above results, we constructed a Lypunov function in terms of drift and loss for some $\theta >0$, and prove the following bound by appropriately choosing the learning rate $\eta$ and constants:
> $$\Phi\left(\underline{\mathbf w}^{r+1}\right) + \theta \mathcal{D}\_{r+1,0} \leq \left(1-\frac{\eta\mu}{16} \right)\left(\Phi\left(\underline{\mathbf w}^{r}\right) + \theta \mathcal{D}\_{r,0}\right).$$
>
> The above shows that the drift and the loss go down to zero exponentially fast (linear convergence). The above approach is quite different from [R2]. In summary, the authors in [R2] were able to exploit the existence of unique minima and strong convexity to prove the result. Whereas in our setting the local functions can converge to different optimal points which makes our analysis challenging and different from [R2].

---

> > ### Author Response · Authors · 2024-07-31
> > **Response to Reviewer ztBN (Part 3)**
> >
> > > **Your Comment:** Clarify the situation:
> >
> > > R3. [Distributed optimization for overparameterized problems: Achieving optimal dimension independent communication complexity.] gives linear convergence of deterministic decentralised GD under the interpolation and PL assumptions.
> >
> > **Our Response:**  The authors in [R3] consider a **deterministic setting** as opposed to the **stochastic setting** considered in our case. It is well known that for a deterministic setting GD-based algorithms achieve linear convergence. Moreover, the algorithms proposed in [R3] are not **federated** in the sense that the local nodes in the proposed algorithms conduct only a single local update. In contrast, the setting considered in our work is significantly challenging stochastic setting with a federated setup where each agent conducts multiple **local updates** within each communication round. We would like to point out that unlike the deterministic setting, it is not clear if under this setting linear convergence can still be achieved. Specifically, the local updates combined with the stochastic setting raises new challenges in the analysis of the algorithms, especially, without imposing any heterogeneity assumptions. We would like to emphasize that these challenges are non-trivial and have not been addressed in the past.
> >
> >
> > In addition to the above, the emphasis of [R3] is more on quadratic loss and considers neural network with **mean squared error loss**, which is very restrictive. In contrast, we consider any system (including neural network) with loss satisfying PL-inequality. Therefore, our work not only is different in terms of analysis but also provides new results in case of general loss functions with multiple local rounds.
> >
> >
> > In summary, we believe that the analyses considered in [R1], [R2] and [R3] combined are different from our work, and our result does not follow directly from their work.
> >
> > > **Your Comment:** The current paper gives linear convergence of decentralised SGD under the interpolation and PL assumptions. In addition, the paper includes the scenario where the nodes communicate after T local iterations (but this is usually quite trivial).
> >
> >
> > **Our Response:** We thank the reviewer for the comment. As pointed by the reviewer we consider the setting where the nodes communicate after T local iterations, which leads to a phenomenon termed as "client drift".  This causes the local models drift apart from each other because of heterogeneity in the local datasets. In many standard decentralised analyses this drift is controlled using the bounded heterogeneity (or gradient) assumptions. Since we do not assume bounded heterogeneity, bounding the drift term (due to multiple local rounds $T \neq 1$) poses additional challenges.
> >
> > As a consequence of the execution of local updates within each communication round, the nodes do not have consensus. This implies that we need to control the consensus error in addition to the client drift (this is absent in [R1] and [R3]). We carefully bound the client drift in terms of the consensus error and global loss by using the smoothness assumption. In addition, we use the recursion on the local loss function to obtain global loss. The next challenge lies in establishing recursion on the drift term, which poses similar challenges to that of proving bound on the loss. Finally, the two recursions (bounds) are combined carefully to show that the sum goes down to zero exponentially fast.
> >
> > > **Your Comment:** The paper also claims to have removed some standard assumptions on bounded variance, heterogeneity etc., but I don't see any of them in [The power of interpolation: Understanding the effectiveness of sgd in modern over-parametrized learning]. Could the authors comment on that?
> >
> > **Our Response:** We thank the reviewer for the comment. We agree with the reviewer that the authors in [R1] do not consider bounded variance assumption. In case of [R1], bounded heterogeneity assumption is not required as the paper considers a single-node centralised setting. We note that bounded heterogeneity assumption is usually required to bound the client drift occurring because of local updates at each client. This is a major contribution of our work where we consider the decentralised SGD setting with multiple local rounds and still are able show linear convergence without the bounded heterogeneity assumption.
> >
> >
> > > **Your Comment:** I would like the authors to clarify my questions and modify accordingly the introduction and related work sections in order to make the contribution clear.
> >
> > **Our Response:** We thank the reviewer for the comment. We hope that the responses to the above comments have clarified our contributions. In the updated manuscript we will modify the introduction and related works to clearly indicate our contributions.

---

> > > ### Author Response · Authors · 2024-08-13
> > > **Request for Response**
> > >
> > > We sincerely thank the reviewer for the time, effort, and valuable feedback. We have uploaded our response to OpenReview, including a detailed address of each comment. Since the rebuttal phase is coming to an end, we were hoping to get a response from you regarding any concerns you may have.
> > >
> > > We will be happy to address them.
> > >
> > > Regards, Authors

---

### Review · Reviewer_7fXb · 2024-08-01

**Summary Of Contributions:**

The paper studies a decentralized federated learning problem where the population objective satisfies the Polyak--Łojasiewicz condition and the individual losses are zero at the population-level optimum (so that no noise at optimum). This corresponds to the ``interpolation regime'' in modern ML models. The paper analyzed decentralied FedAvg algorithm where each device runs a mini-batched SGD algorithm and the information is exchanged every $T$ rounds through a mixing matrix. The paper proved linear convergence with an outer-loop iteration complexity of order $O(T^2 \log (1 / \varepsilon))$, where $T$ is the number of steps in an inner loop and $\varepsilon$ is the desired accuracy level.

**Audience:**

Yes

**Broader Impact Concerns:**

I don't see any potential ethical concerns or broader impact issues.

**Claims And Evidence:**

No

**Requested Changes:**

The writing needs significant improvements. For example, Theorem 1 has 12 stepsize conditions involving a lot of messy notations. It is not clear which condition is relevant. It would be much better to simplify it significantly by considering the usual scaling of these parameters and removing the irrelevant conditions. Also, it would be better to state the main theorem in the form of final complexity guarantees instead of per-iteration progress.

Here are some minor comments:
- There are lots of notation conflicts. For example, lemmas indexed as Lemma 1 and Lemma 2 appeared on both page 5 and page 8, and they are completely different lemmas.
- Table 1: I don't understand ``SC, C and NC represent Server, Decentralized, Strongly
convex, Convex and Non-convex settings, respectively.''. It seems that ``Server, Decentralized'' should not be here.

**Strengths And Weaknesses:**

Strength. Compared to existing literature, the paper extends the analysis in Koloskova et al., (2020) from strongly convex settings to non-convex functions satisfying Polyak--Łojasiewicz condition; compared to the existing work by Song et al., (2022) which also dealt with the PL conditions, the paper allows mini-batched SGD on each machine, as opposed to full gradients. The results are technically new.

Weakness. Given a lot of existing works in this literature, the novelty of this paper is relatively limited. The relaxation from strongly convex functions to PL conditions is well-troden path in optimization literature. Furthermore, there are already existing works under the PL conditions for deterministic setups. In the interpolation regime, extension from deterministic to stochastic settings while keeping the linear convergence requires some technicalities, but is not very surprising.

Moreover, the results in this paper are weak compared to many existing works. The number of communication rounds needs to scale as $O(T^2 \log (1 / \varepsilon)/ \mu)$, which is worse than prior papers in the strongly convex settings. This slowdown is reflected in the $O (1/T^2)$ stepsize requirement in Theorem 1. As the length of inner loops grow larger, the actual move it can make becomes smaller.

It is not clear to me whether the stepsize choice in Corollary 2 is always attainable. In particular, Theorem 1 requires that $\eta \leq \frac{\mu}{\zeta_8 T^2}$, while Corollary 2 sets $\eta = \frac{\mu}{(\mu^2 + 8) T}$. Comparing the two facts yields $T \leq \frac{\mu^2 + 8}{\zeta_8}$. Suppose that $\mu$ is $O(1)$ and the smoothness constants etc are large, we may have $\frac{\mu^2 + 8}{\zeta_8} < 1$, and there is no possitive integer $T$ satisfying this relation.

---

> ### Author Response · Authors · 2024-08-06
> **Response to Reviewer 7fXb (Part 1)**
>
> > **Your Comment:** Given a lot of existing works in this literature, the novelty of this paper is relatively limited. The relaxation from strongly convex functions to PL conditions is well-troden path in optimization literature. Furthermore, there are already existing works under the PL conditions for deterministic setups. In the interpolation regime, extension from deterministic to stochastic settings while keeping the linear convergence requires some technicalities, but is not very surprising.
>
> **Our Response:** We thank the reviewer for the comment. The statement "relaxation from strongly convex functions to PL conditions is well-troden path in optimization literature" is in general true for a single node setting however the distributed setting poses several challenges compared to the single node setting, as explained next.
>
> In the case of distributed setting for the strongly-convex objectives the local clients share a unique minimum, which allows the algorithm to converge with the same rate to the local optimal irrespective of the local updates and communication protocol used. In contrast, for the PL functions the local functions can converge to different (heterogeneous) local optimum points. This makes the analysis more challenging compared to the simple strongly-convex setting.  Further, it is well known that in the deterministic setting distributed algorithms for minimizing PL loss functions are capable of achieving linear convergence (even in the non-interpolated regime). Moreover, since deterministic problems involve computations of very large batch gradients during each update, they are not practical, especially, for large scale problems. In contrast, our setting considers a stochastic problem which is significantly more challenging compared to a deterministic setting, since we have to deal with stochasticity of the algorithm at each local update. Moreover, it is an open problem to establish if (stochastic) decentralized FedAvg can guarantee linear convergence in the interpolation regime. In addition, the stochastic setting allows us to compute the small batch gradients, that makes the considered algorithms computationally more practical.
>
>
> > **Your Comment:** Moreover, the results in this paper are weak compared to many existing works. The number of communication rounds needs to scale as $\mathcal{O}({T^2} \log ({1}/{\epsilon})/\mu)$, which is worse than prior papers in the strongly convex settings. This slowdown is reflected in the $\mathcal{O}(1/T^2)$ stepsize requirement in Theorem 1. As the length of inner loops grow larger, the actual move it can make becomes smaller.
>
> **Our Response:** We thank the reviewer for the comment. We agree with the reviewer that in our case the number of communication rounds needs to scale as $\mathcal{O}({T^2} \log ({1}/{\epsilon})/\mu)$. We believe that the slightly worse convergence of our analysis is due to the non-convex functions satisfying the PL inequality. Importantly, as a special case of our analysis, for the classical FedAvg algorithm (with a server), we establish a convergence rate of $\mathcal{O}(\log({1}/{\epsilon}))$ which is significantly better than that existing results. Moreover, via our analysis we capture the effect of local updates on the performance of the decentralized FedAvg. Specifically, we show that there exists an optimal number of local updates $T$ beyond which the convergence of the algorithm slows down and one needs to choose $T$ carefully to achieve the optimal convergence guarantees.

---

> ### Author Response · Authors · 2024-08-06
> **Response to Reviewer 7fXb (Part 2)**
>
> > **Your Comment:** It is not clear to me whether the stepsize choice in Corollary 2 is always attainable. In particular, Theorem 1 requires that $\eta \leq \frac{\mu}{\zeta_8 T^2 }$, while Corollary 2 sets $\eta = \frac{\mu}{\left(\mu^2+8\right)T}$. Comparing the two facts yields $T \leq \frac{\mu^2+8}{\zeta_8}$. Suppose that $\mu$ is $\mathcal{O}(1)$ and the smoothness constants etc are large, we may have $\frac{\mu^2+8}{\zeta_8} \leq 1$, and there is no positive integer $T$ satisfying this relation.
>
> **Our Response:** We thank the reviewer for the comment. We apologize for the lack of clarity in the presentation. We agree that Theorem 1 requires $\eta \leq \frac{\mu}{\zeta_8 T^2 }$ which holds for any number of local rounds $T$, and is applicable for decentralized setting. Note that Corollary 2 is a separate result and only holds for the classical FedAvg (with a server, i.e., $\lambda_2 = 0$) which is a special case of the decentralized setting. Specifically, we do a refined analysis of the classical FedAvg, as explained next. Recall, the following bound on the loss function from equation 43 of Sec. C.1
>
> $$\mathbb{E}\left[\Phi\left(\underline{\mathbf w}^{r+1}\right)\right] \leq \mathbb{E}\left[\left(\left(1-\frac{\eta \mu}{4}\right)^T+\frac{64 \eta^{4}  T^{3} l\_{max} L L\_{max}}{\mu\_{min}}\right) \Phi\left(\underline{\mathbf w}^{r}\right) + \frac{4\eta  TL^{2}}{N} \lVert(Q-P) W^{r, 0}\rVert_{F}^{2} +\right.$$
> $$\left.\frac{2\eta  TL^{2}}{N}\left[\left(\frac{16 l\_{max}\eta^{2}  T^{2} L\_{max}^{2}}{\mu\_{min}}+4 \lambda\_{2}^{2} \eta^{2} \gamma L\_{max}^{2}  T^{2}\right)\mathcal{D}\_{r,0} + 4 \eta^{2} \gamma  T^{2} \lambda\_2^2 \lVert\partial \Phi\left(\underline{W}^{r, 0}\right)\rVert_{F}^{2}\right]\right]. $$
>
> By setting $\lambda_2 =0$ (classical FedAvg) in the above, we get
> $$\mathbb{E}\left[\Phi\left(\underline{\mathbf w}^{r+1}\right)\right] \leq  \left(\left(1-\frac{\eta \mu}{4}\right)^T+\frac{64 \eta^{3}  T^{3} l\_{max} L^2 L\_{max}}{\mu\_{min}}\right) \mathbb{E}\left[\Phi\left(\underline{\mathbf w}^{r}\right)\right], $$
>
> Now, optimizing the above with respect to $T$, we get $T_{th} = \frac{\mu}{\eta\left(\mu^2+8\right)}$. Further, $\eta = \frac{\mu}{T_{th}\left(\mu^2+8\right)}$ in Corollary 2 holds for a particular value of $T$, i.e., when $T = T_{th}$. Hence comparing the two results is not appropriate. Once again, we apologize for the confusion.
>
>
>
> > **Your Comment:** The writing needs significant improvements. For example, Theorem 1 has 12 stepsize conditions involving a lot of messy notations. It is not clear which condition is relevant. It would be much better to simplify it significantly by considering the usual scaling of these parameters and removing the irrelevant conditions. Also, it would be better to state the main theorem in the form of final complexity guarantees instead of per-iteration progress.
>
> **Our Response:** We apologize to the reviewer for the lack of clarity in presenting the theorem. In the updated manuscript, we have restated the theorem as follows:
>
> **Theorem 1** Under Assumptions 1-5, after $T$ local iterations, choosing $\eta = \mathcal{O}(1/T^2)$, we get
> $$\mathbb{E}[\Phi\left(\underline{\mathbf w}^{r+1}\right) + \theta \mathcal{D}_{r+1,0}] \leq \left(1-\frac{\eta \mu}{16} \right)\mathbb{E}[\Phi\left(\underline{\mathbf w}^{r}\right) + \theta \mathcal{D}\_{r,0} ]$$
>
>
> In summary, our results do not follow directly from the strongly convex setting. The technicalities involved in proving the result are significantly different compared to the strongly convex setting. Moreover, the existing results on dectralized setting with PL do not carry out convergence analysis for a general loss function with multiple local updates as they consider only quadratic loss with multiple layer neural network. Multiple local rounds without assuming bounded heterogeneity is difficult to handle. In addition, our results shed new light on the convergence of vanilla FedAvg algorithm.
>
> > **Your Comment:** Here are some minor comments:
>
> - There are lots of notation conflicts. For example, lemmas indexed as Lemma 1 and Lemma 2 appeared on both page 5 and page 8, and they are completely different lemmas.
> - Table 1: I don't understand SC, C and NC represent Server, Decentralized, Strongly convex, Convex and Non-convex settings, respectively.''. It seems that Server, Decentralized'' should not be here.
>
> **Our Response:** We apologize to the reviewer for the typos. In the updated manuscript we have corrected the typos.

---

> > ### Author Response · Authors · 2024-08-13
> > **Request for Response**
> >
> > We sincerely thank the reviewer for the time, effort, and valuable feedback. We have uploaded our response to OpenReview, including a detailed address of each comment.  Since the rebuttal phase is coming to an end, we were hoping to get a response from you regarding any concerns you may have.
> >
> > We will be happy to address them.
> >
> > Regards,
> > Authors

---

### Review · Reviewer_pU3c · 2024-09-05

**Summary Of Contributions:**

FedAvg is a well-known algorithm for federated learning. While it has been studied in the centralized setting, the convergence of FedAvg in the decentralized setting is not clear. This paper makes an important contribution to exploiting the convergence of decentralized FedAvg under the PL condition.

**Audience:**

Yes

**Broader Impact Concerns:**

Not available.

**Claims And Evidence:**

Yes

**Requested Changes:**

The comparison between underparameterized and overparameterized settings seems a bit distracting, as it primarily shows the advantages of overparameterized models, which is known before. I am okay if you keep it, but since you are working with the PL setting and linear convergence is expected, could you plot the convergence of the optimality gap f(w^t) - f(w^*), where w^* is an approximate optimal solution? For instance, you could run any centralized algorithm for a sufficient amount of time to obtain this. So far, all the losses seem to be sublinear.

**Strengths And Weaknesses:**

Strengths

- The paper provides a new analysis of Decentralized FedAvg, focusing on the interpolation regime and non-convex functions satisfying the PL inequality. The federated optimization under PL seems to be a relatively unexplored area, and the results have significant implications for decentralized learning.

Weakness

- The convergence analysis under the PL setting seems to be quite standard, can you highlight any technical novelty in case i missed something?

- Is there any stronger baseline algorithm that you can compare with?

---

> ### Author Response · Authors · 2024-09-11
> **Response to Reviewer pU3c (Part 1)**
>
> > **Your Comment:** The convergence analysis under the PL setting seems to be quite standard, can you highlight any technical novelty in case i missed something?
>
> **Our Response:** We agree with the reviewer that the convergence analysis under the PL setting is well known under the centralized setting. But the decentralized setting poses several challenges. Note that in the decentralized setting considered in our work the individual nodes execute multiple local updates. We would first like to point out that the local updates in conjunction with a decentralized topology have received relatively lesser attention by the research community partly because of the challenging nature of the problem. In the decentralized setting, the "client drift" arising because of multiple local updates and the consensus error because of the absence of the server creates unique challenges in the analysis of FedAvg-type algorithms. Moreover, it is unknown what kind of convergence guarantees can be achieved in such a setting. Towards handling these challenges we prove the following bounds on the loss in terms of drift, $\mathcal{D}\_{r,0} := \sum_{k=1}^{N}\mathbb{E}\lvert|\underline{\mathbf w}_{k}^{r,0}-\underline{\mathbf w}^{r,0}\rvert|^{2}$, and vice versa as shown in the following theorems.
>
>
> **Theorem:** The expected loss is bounded in terms of the drift as follows
> $$\mathbb{E}\left[\Phi\left(\underline{\mathbf w}^{r+1}\right)\right] \leq \left(1-\frac{\eta \mu}{8}\right) \Phi\left(\underline{\mathbf w}^{r}\right)+\frac{6 \eta L^2T}{N} \mathcal{D}_{r,0}.$$
>
> **Theorem:** The drift is bounded in terms of expected loss as follows
> $$\mathcal{D}\_{r,0} \leq
> \left(\left(1+\frac{1}{\psi}\right)\lambda\_2^2 + \eta^{2} \lambda\_{2}^{4}\beta T^2L\_{max}^2\right)\mathcal{D}\_{r-1,0} + 2\eta^{2} \lambda\_{2}^{4}\beta T^2 L\_{max}N \mathbb{E}\left[\Phi \left(\underline{\mathbf w}^{r-1, 0}\right)\right]$$
>
> The parameters above are defined in the paper. Note that proving the above bound involves carefully handling the drift due to multiple local updates. Using the above results, we constructed a Lypunov function in terms of drift and loss for some $\theta >0$, and prove the following bound by appropriately choosing the learning rate $\eta$ and constants:
> $$\Phi\left(\underline{\mathbf w}^{r+1}\right) + \theta \mathcal{D}\_{r+1,0} \leq \left(1-\frac{\eta\mu}{16} \right)\left(\Phi\left(\underline{\mathbf w}^{r}\right) + \theta \mathcal{D}\_{r,0}\right).$$
>
> The above shows that the drift and the loss diminish to zero exponentially fast (linear convergence). We note that this is the first result to establish the linear convergence of decentralized FedAvg for minimizing PL objectives in the interpolation regime.
>
> > **Your Comment:** Is there any stronger baseline algorithm that you can compare with?
>
> **Our Response:** We thank the reviewer for the comment. We are not sure if we have understood the comment correctly. We assume that the reviewer is asking for comparison with other existing decentralized algorithms. FedAvg is a standard algorithm both in the centralized setting where the server coordinates the information sharing among clients as well as the decentralized setting where the neighbouring nodes communicate based on network topology. However, there are several algorithms in the decentralized setting such as momentum SGD. In the updated manuscript we have included the comparison of decentralized SGD with momentum (DSGDm) and the decentralized FedAvg algorithm. As expected, DSGDm outperforms the FedAvg algorithm. For a quick access, we have reproduced the results in Table 1 below:
>
> | $R$ | Training loss (DSGDm)|Training loss (FedAvg)|
> |:---:|:-----------------:|:--------------------:|
> | 20  |      0.160979      | 0.243191             |
> | 50  |      0.023768      | 0.081793            |
> | 100 |      0.005945      | 0.026803             |
> | 150 |      0.002603      | 0.015210            |
> | 200 |      0.001393      |0.009380            |
> | 250 |      0.000852      | 0.006262           |
>
>
> Table 1: Comparison with existing algorithms

---

> > ### Author Response · Authors · 2024-09-11
> > **Response to Reviewer pU3c (Part 2)**
> >
> > > **Your Comment:** The comparison between underparameterized and overparameterized settings seems a bit distracting, as it primarily shows the advantages of overparameterized models, which is known before. I am okay if you keep it, but since you are working with the PL setting and linear convergence is expected, could you plot the convergence of the optimality gap f(w^t) - f(w^*), where w^* is an approximate optimal solution? For instance, you could run any centralized algorithm for a sufficient amount of time to obtain this. So far, all the losses seem to be sublinear.
> >
> > **Our Response:** We thank the reviewer for the comment. According to the reviewer's suggestion, in the updated manuscript we have included the plot of optimality gap versus the communication rounds ($R$). As expected the optimality gap decreases exponentially with $R$. For a quick access we have reproduced the results in Table 2:
> >
> >
> > | $R$ | Optimality gap  $$\Phi\left(\underline{\mathbf w}^{r}\right) - \Phi\left(\underline{\mathbf w}^{*}\right)$$|
> > |:---:|:-----------------:|
> > | 50  |      0.121086      |
> > | 100 |      0.094527     |
> > | 200 |      0.084278      |
> > | 250 |      0.079226       |
> > | 300 |      0.07212       |
> > | 400 |      0.064104       |
> >
> > Table 2: Effect of the optimality gap

---

### Decision · Action_Editor_Jh46 · 2024-11-26

**Recommendation:** Accept as is

**Comment:**

The responses from the authors helped to clarify a few points and also led to improved presentation of the work. The main criticism is the incremental nature of the work, as it puts together several aspects that have been well studied separately (decentralized learning in the interpolation regime, for PL objectives, with local updates). Yet, combining these aspects is relevant and overall, the wok satisfies TMLR's criteria.

**Audience:**

The paper is about non-convex federated learning, so will attract sufficiently broad audience.

**Claims And Evidence:**

Yes, the paper provides proofs of convergence for their approach, as well as numerical experiments. The related work is appropriately cited and the paper is adequately positioned with respect to it.